# DeBoNet: A deep bone suppression model ensemble to improve disease detection in chest radiographs

**Sivaramakrishnan Rajaraman**[1]*, **Gregg Cohen**[2], **Lillian Spear**[2], **Les Folio**[2¤], **Sameer Antani**[1]

**1** National Library of Medicine, National Institutes of Health, Bethesda, Maryland, United States of America, **2** Clinical Center, Department of Radiology and Imaging Sciences, National Institutes of Health, Bethesda, Maryland, United States of America

¤ Current Address: Current affiliation: Moffitt Cancer Center and Research Institute, Tampa, Florida, United States of America

* sivaramakrishnan.rajaraman@nih.gov

**Data Availability Statement:** The minimal data required to reproduce this study are available in terms of the figures, performance metrics, and other measured reported in the tables. The code

## Abstract

Automatic detection of some pulmonary abnormalities using chest X-rays may be impacted adversely due to obscuring by bony structures like the ribs and the clavicles. Automated bone suppression methods would increase soft tissue visibility and enhance automated disease detection. We evaluate this hypothesis using a custom ensemble of convolutional neural network models, which we call DeBoNet, that suppresses bones in frontal CXRs. First, we train and evaluate variants of U-Nets, Feature Pyramid Networks, and other proposed custom models using a private collection of CXR images and their bone-suppressed counterparts. The DeBoNet, constructed using the top-3 performing models, outperformed the individual models in terms of peak signal-to-noise ratio (PSNR) (36.7977±1.6207), multi-scale structural similarity index measure (MS-SSIM) (0.9848±0.0073), and other metrics. Next, the best-performing bone-suppression model is applied to CXR images that are pooled from several sources, showing no abnormality and other findings consistent with COVID-19. The impact of bone suppression is demonstrated by evaluating the gain in performance in detecting pulmonary abnormality consistent with COVID-19 disease. We observe that the model trained on bone-suppressed CXRs (MCC: 0.9645, 95% confidence interval (0.9510, 0.9780)) significantly outperformed ($p < 0.05$) the model trained on non-bone-suppressed images (MCC: 0.7961, 95% confidence interval (0.7667, 0.8255)) in detecting findings consistent with COVID-19 indicating benefits derived from automatic bone suppression on disease classification. The code is available at https://github.com/sivaramakrishnan-rajaraman/Bone-Suppresion-Ensemble.

## Introduction

Chest X-ray (CXR) is a commonly performed radiological examination to visualize various abnormalities in the thoracic cavity [1]. However, accurate interpretation of pulmonary

and related information for the algorithm are made publicly available at https://github.com/sivaramakrishnan-rajaraman/Bone-Suppresion-Ensemble to promote global research.

**Funding:** This research was supported by the Intramural Research Program of the National Library of Medicine, and the Clinical Center, both parts of the National Institutes of Health. The funders had no role in study design, data collection, and analysis, decision to publish, or preparation of the manuscript.

**Competing interests:** The authors have declared that no competing interests exist.

abnormalities like COVID-19 and others is particularly challenging because their visibility may be obstructed by the presence of bony structures like ribs and clavicles. This reduced visibility may lead to an erroneous interpretation by an expert or an artificial intelligence (AI) algorithm, thereby severely impacting clinical decision-making. It has been noted in the literature that the presence of ribs and clavicles in CXR images led to missed lung cancer nodules resulting in false interpretations [2].

Advanced radiology methods like dual-energy subtraction (DES) chest radiography are used to produce "bone-only" and soft tissue images [3]. However, compared to traditional CXRs, DES has several limitations [4]: (i) DES radiography exposes the subjects to a slightly higher radiation dosage compared to traditional CXR imaging; (ii) DES radiographic imaging can only be performed in the posterior-anterior view; (iii) Cost of DES radiography is higher compared to conventional CXR imaging; and, (iv) DES radiography is recommended only for patients above 16 years of age. Therefore, an automated bone suppression method for traditional CXRs should add value for enhancing soft tissue visibility and aid in the improved detection of pulmonary manifestations.

A study of the literature reveals several works published on suppressing bones in CXRs. These studies involve using (i) commercial software, (ii) conventional machine learning methods using hand-crafted feature descriptors, or (iii) state-of-the-art deep learning (DL) models to initially generate bone-only images and further subtract them from the original CXR to increase soft-tissue visibility. In [5], the authors used commercial software to suppress bones and improve performance for detecting lung nodules. It was observed that the performance of the experts significantly improved ($p < 0.05$) by using the bone-suppressed CXRs resulting in an area under the receiver-operating-characteristic curve (AUROC) of 0.863, compared to an AUROC of 0.82 using non-bone-suppressed CXRs. Another study used commercial software to suppress bones in CXRs and investigated for a performance improvement in Tuberculosis (TB) detection [6]. They observed that the average AUROC of experts improved from 0.882 to 0.933 using bone-suppressed images. A convolutional neural network (CNN)-based model was used in [7] to generate a bone-only image. This image is subtracted from the original CXR to increase soft-tissue visibility, thereby resulting in 89.2% bone suppression. A cascade of CNNs was used in [8] to create bone-only images at multiple scales. The generated images were fused to form the final bone-only image that was subtracted from the original CXR to generate a "bone-free" image. In another study [9], an artificial neural network was used to generate a bone-only image that was subtracted from the original CXR to increase the visibility of soft tissues. A method based on independent component analysis was proposed in [10] to suppress bones and increase lung nodule visibility. Other studies [11–14] adopted bone suppression methods to improve performance toward detecting lung nodules and other pulmonary manifestations. These studies in general, propose multiple steps to generate bone-only images and subtract them from the original CXRs to increase soft-tissue visibility. A limitation of this approach is that an inaccurate generation of bone-only images would lead to introducing noise, reducing the visibility of soft tissues, increasing interpretation errors, and adversely impacting decision-making. As of the writing of this manuscript and to the best of our knowledge, other than [15] there are no other articles in the literature that propose an automated method to generate a soft-tissue image directly from the original CXR image, alleviating the need for intermediate bone image generation and subsequent subtraction methods.

Though CNN models demonstrate state-of-the-art performance in natural and medical vision recognition tasks, they are often found to suffer from bias and variance issues that could adversely affect their interpretation. These issues could be tackled through ensemble learning that optimally combines the predictions of several models to improve prediction performance compared to the individual constituent models and reduce prediction spread or dispersion

[16]. Ensemble learning is widely used in medical computer vision tasks such as segmentation, object detection, and classification [17]. To the best of our knowledge, we do not find any literature that evaluates the performance of DL model ensembles for bone suppression in CXRs.

In this study, we propose DeBoNet, an ensemble of DL models, for suppressing bones in frontal CXRs. Through its use, we aim to improve disease classification and interpretation performance which is demonstrated through the detection of findings that are consistent with COVID-19 on CXRs [18]. We train several state-of-the-art architectures such as U-Nets [19] and Feature Pyramid Networks (FPNs) [20], using several ImageNet classifiers as backbones, and also propose custom models toward bone suppression. DeBoNet is constructed by (i) measuring the multi-scale structural similarity index (MS-SSIM) score between the sub-blocks of the bone-suppressed image predicted by each of the top-3 performing bone-suppression models and the corresponding sub-blocks in the respective ground truth soft-tissue image, and (ii) performing a majority voting of the MS-SSIM score computed in each sub-block to identify the sub-block with the maximum MS-SSIM score and use it in constructing the final bone-suppressed image. We empirically determine the sub-block size that delivers superior bone suppression performance. The performances of individual models and DeBoNet are evaluated using several performance metrics such as average peak signal-to-noise (PSNR) ratio, structural similarity index (SSIM), MS-SSIM, correlation, intersection, chi-square, and Bhattacharya distances. Next, the best-performing bone suppression model is selected, truncated, and appended with classification layers. This is done to transfer CXR modality-specific knowledge and improve performance in the task of classifying CXRs as showing normal lungs or other findings consistent with COVID-19. The performance of the classification model trained on non-bone-suppressed CXRs and bone-suppressed CXRs are compared through several performance metrics such as accuracy, AUROC, precision, recall, the area under the precision-recall curve (AUPRC), F-score, and MCC. Additionally, we used our in-house class-selective relevance map (CRM) algorithm [21] to interpret model predictions. Fig 1 shows the graphical abstract of our proposed approach.

Our novel contributions are highlighted as follows:

(i) To the best of our knowledge, this is the first study to develop a model ensemble for suppressing bones in CXRs, that we call DeBoNet, and demonstrate its effectiveness through extensive qualitative and quantitative analyses.

(ii) We train and evaluate variants of U-Nets, Feature Pyramid Networks, and other proposed custom models toward the bone suppression task.

(iii) The individual constituent models and the DeBoNet proposed in this study are not restricted to the task of CXR bone suppression but can be potentially applied to other image denoising applications.

## Materials and methods

### Datasets

The following datasets are used in this study:

(i) COVID-19 CXR collection: A total of 3016 de-identified publicly available CXR images showing findings that are consistent with COVID-19, which serve as the set of cases in our study, are pooled from several sources. A majority of these CXRs are pooled from the BIMCV-COVID19+ CXR data collection that contains 2473 CXRs showing COVID-19-like manifestations [22]. A set of 183 CXR images showing findings consistent with COVID-19 are collected from a GitHub repository hosted by the Institute for Diagnostic

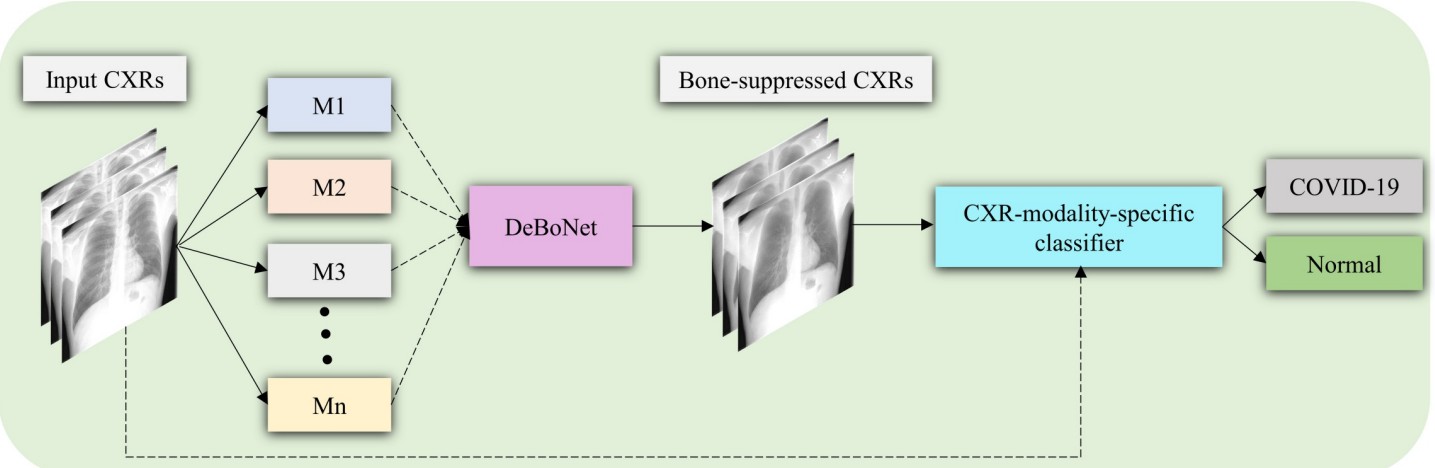

**Fig 1. Graphical abstract of the proposal.** Several proposed bone suppression models (M1, M2, . . ., M$n$, $n$ = 1, 2, . . ., 14) are trained on a set of input CXRs. The predictions of the top-3 performing bone suppression models are combined using a majority voting approach to construct DeBoNet. A classification model is trained on the non-bone-suppressed and bone-suppressed images to classify them into COVID-19 or normal categories.

and Interventional Radiology, Hannover Medical School, Hannover, Germany [23]. These CXR images are accompanied by other metadata such as admission status and patient demographics. The authors [17] collected 226 CXRs manifesting COVID-19, from a public GitHub repository hosted by the authors of [24]. The CXR collection is accompanied by other metadata including sex, age, finding, and intubation status. They also used a collection of 134 CXRs acquired from SARS-CoV-2 PCR+ patients from a hospital in Spain and posted by a radiologist in a public Twitter thread [25]. The ground truth COVID-19 disease-specific region of interest (ROI) annotations, set by the verification from two expert radiologists, for a subset of this collection [$n$ = 36] are used by the authors of [16] in interpreting model performance.

(ii) RSNA CXR dataset: To serve as experimental controls, we randomly select an equal number of 3016 de-identified CXR images showing no abnormalities from the publicly available RSNA CXR dataset, released toward the RSNA pneumonia detection challenge hosted by Kaggle [26]. The collection, however, includes a total of 26,684 CXR images, of which, 8851 CXRs showed no abnormalities, 6012 CXRs showed pneumonia-related lung opacities, and 11,821 CXRs showed other pulmonary abnormalities.

(iii) NIH-CC-DES-Set 1: This set consists of 27 de-identified DES CXR images [15] that were acquired at the National Institutes of Health (NIH) Clinical Center (CC) as a part of routine clinical care. A GE Discovery XR656 digital radiography system was used to acquire the DES images at 120 and 50 Kilovoltage-peak (kVp), respectively, to capture the soft-tissue images and bone-only images. This dataset is used to evaluate the performance of the bone suppression models proposed in this study.

(iv) NIH-CC-DES-Set 2: Another set of de-identified 100 DES CXRs are acquired similar to NIH-CC-DES-Set 1. This collection contains DES images of 54 females and 46 males, the average age and standard deviation of the males and females are 48.9 +/- 14.5 and 45.4+/- 13.6, respectively. This dataset is augmented and used to train the bone suppression models.

The NIH-CC-DES-Set 1 and NIH-CC-DES-Set 2 data were selected samples of adult subjects with no radiological findings from the NIH archives that were deidentified and manually

verified before use. The NIH Institutional Review Board (IRB) exempted their use from full review. The total number of CXRs pooled from different sources is given in Table 1.

## Bone suppression models

The set of 100 grayscale DES CXR images (i.e., the original CXRs and soft tissue counterparts) from the NIH-CC-DES-Set 2 dataset is augmented using affine transformations such as rotations (-10 to 10 degrees), horizontal and vertical shifting (-5 to 5 pixels), horizontal mirroring, zooming, median filtering, Gaussian blurring, and unsharp masking, resulting in 1000 DES CXRs. The augmented images are further resized to 256×256 dimensions to reduce computational complexity. The contrast of the images is enhanced by clipping the top and bottom 1%, respectively, of all pixel values. The pixel values are then normalized.

We propose the following model architectures for the task of bone suppression in CXRs: (i) Autoencoder-BS (BS—Bone Suppression); (ii) ResNet-BS; (iii) U-EB0-BS; (iv) U-Res18-BS; (v) U-SE-Res18-BS; (vi) U-D121-BS; (vii) U-IV3-BS; (viii) U-MobileV2-BS; (ix) F-EB0-BS; (x) F-Res18-BS; (xi) F-SE-Res18-BS; (xii) F-D121-BS, (xiii) F-IV3-BS; (xiv) F-MobileV2-BS. These model architectures are discussed in subsequent sections.

**Autoencoder with separable convolutions (Autoencoder-BS).** The Autoencoder-BS model is a denoising autoencoder with symmetrical encoder and decoder layers. Fig 2 illustrates the architecture of the proposed Autoencoder-BS model.

The encoder consists of four separable convolutional blocks. Each convolutional block except for the last block contains two separable convolutional layers. Separable convolutions are used to reduce computational complexity, thereby facilitating faster convergence and real-time deployment [27]. The number of filters in the separable convolutional blocks of the encoder are 64, 128, 256, and 512, respectively. Except for the last block, a max-pooling layer is used after each separable convolutional block to calculate the maximum value for individual patches of the feature map. Upsampling layers are used correspondingly in the symmetric decoder blocks to preserve the spatial resolution of the input.

**ResNet-based model with residual scaling (ResNet-BS).** The architecture of the proposed ResNet-BS model is shown in Fig 3. The first and last convolutional layer contains 128 filters of dimension 3×3. We used residual blocks with shortcuts to skip over layers. This approach helps to overcome convergence issues due to vanishing gradients in deeper models. Skipping layers helps to reuse the activations of the earlier layers until weight updates in the succeeding layers. Each residual block consists of two convolutional layers with 3×3 filters and 128 feature maps.

Inspired by [28], we used a modified residual block in which (i) the batch normalization layers are removed for they are mentioned to adversely affect the range flexibility through the

**Table 1. Dataset sources.**

| Source | Number of CXR images | |
|---|---|---|
| | COVID-19 | Normal |
| BIMCV-COVID19+ CXR | 2473 | No |
| Hannover Medical School, Hannover | 183 | No |
| Cohen et al. | 226 | No |
| Twitter COVID-19 CXR | 134 | No |
| RSNA CXR | 3016 | 3016 |
| NIH-CC—DES-Set 1 | No | 27 |
| NIH-CC—DES-Set 2 | No | 100 |

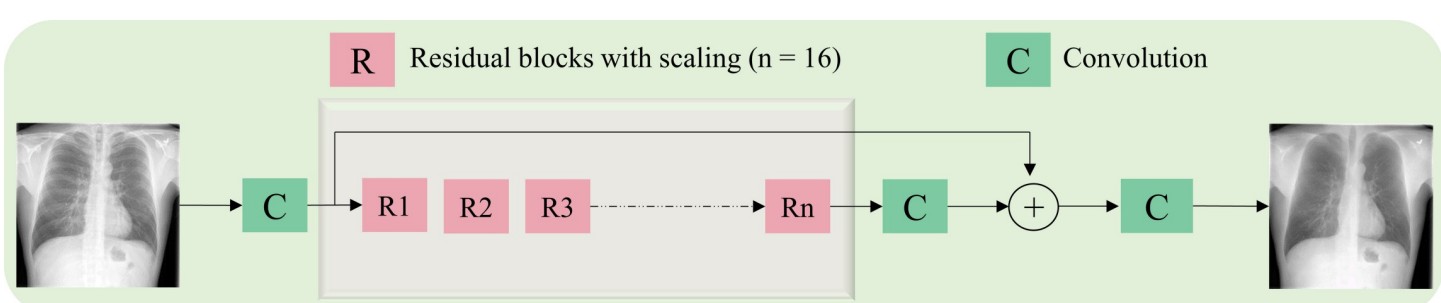

**Fig 2. The architecture of the Autoencoder-BS model.** The input to the model is a grayscale CXR image. The model has a symmetrical separable convolutional encoder and decoder architecture.

normalization process, and (ii) activations are not used outside the residual blocks and in the final layer. The network consists of 16 residual blocks with an identical layout. We used zero paddings to preserve the spatial dimensions of the input image. The residuals after the deepest

**Fig 3. The architecture of the ResNet-BS model.** The input is a grayscale CXR image. The block *R* denotes the modified residual block. The proposed model has 16 residual blocks, each having two convolutional layers with 3×3 filters and 128 feature maps. The deepest convolutional layer with sigmoidal activation predicts the grayscale bone-suppressed image.

convolutional layer in each residual block are scaled at an empirically determined scaling factor (0.1) before adding them back to the convolutional path. This scaling approach stabilizes training in deeper models with high computational complexity [28]. The deepest convolutional layer with the sigmoidal activation function predicts a grayscale bone-suppressed image.

**U-Net and FPN-based models.** The U-Net models are widely used in image segmentation tasks [19]. The U-Net is composed of an encoder and decoder. The encoder or the contracting path extracts image features at multiple scales and the decoder or the expanding path semantically projects the features learned by the encoder onto the pixel space.

The Feature Pyramid Networks (FPN) are widely used as feature extractors to help object detection [20]. Fig 4 shows the general architecture of the U-Net and FPN models. The FPN network is composed of bottom-up and top-down pathways. The bottom-up pathway constitutes the encoder backbone that extracts image features at multiple scales (scaling step is 2). A convolutional layer with a 1×1 filter is used to reduce the feature dimensions of the deepest convolutional layer in the bottom-up pathway to 256. This constitutes the first layer of the top-down pathway. Going deeper, the preceding layer is up-sampled by a factor of 2 using the nearest neighbor up-sampling method. A 1×1 convolutional filter is applied to the corresponding feature maps in the bottom-up pathway and is added elementwise. A 3×3 convolution is then applied to all the merged layers to reduce aliasing effects. This helps to generate high-resolution features at each scale.

The grayscale CXR is duplicated in three channels and fed into the U-Net and FPN models. This is because we use ImageNet-pretrained models, trained on RGB images, as the encoder backbones. We experimented with several encoder backbones for the U-Net and FPN models [29] toward the task of bone suppression in CXRs. These backbones include (i) EfficientNet-B0 [30], (ii) ResNet-18 [31], (iii) SE-ResNet-18 [32], (iv) DenseNet-121 [33], (v) Inception-V3 [34], and (vi) MobileNet-V2 [35]. We are motivated by the fact that these ImageNet-pretrained models have demonstrated superior performance in medical visual recognition tasks [17]. The final layer of the U-Net and FPN models consists of a convolutional layer with Sigmoidal activation to predict grayscale bone-suppressed CXRs.

The proposed bone-suppression models are trained on the augmented NIH-CC-DES-Set 2 dataset and tested with the NIH-CC-DES-Set 1 dataset. We allocated 10% of the training data for validation using a fixed seed. We compiled the models using an Adam optimizer with an initial learning rate of 1e-3 and monitored the following validation performance metrics: (i) loss, (ii) PSNR, (iii) SSIM, and (iv) MS-SSIM. We propose a mixed-loss function that benefits from the combination of mean absolute error (MAE) and MS-SSIM losses, given by,

$$Mixed\ loss = \Omega.MS-SSIM + (1-\Omega).MAE \tag{1}$$

We empirically set the value of $\Omega$ to 0.84. The MS-SSIM metric is given higher weightage since the bone suppressed image is preferred to closely match the ground truth. The MAE metric is given a comparatively lower significance because the metric focuses on the contrast and luminance that is expected to change while suppressing the bones. We reduced the learning rate whenever the validation performance ceased to improve. Early stopping with the patience of 10 epochs is used. The best-performing models (with the least validation loss) are further used to predict bone-suppressed CXR images using the test set. An Ubuntu Linux system with NVIDIA GeForce GTX 1080 graphics card and Keras framework with Tensorflow backend is used for model training and evaluation.

**DeBoNet—bone suppression model ensemble.** The bone suppression model ensemble, which we call DeBoNet, is constructed using the top-3 performing models that demonstrate markedly improved performance in terms of the MS-SSIM metric using the NIH-CC-DES-Set

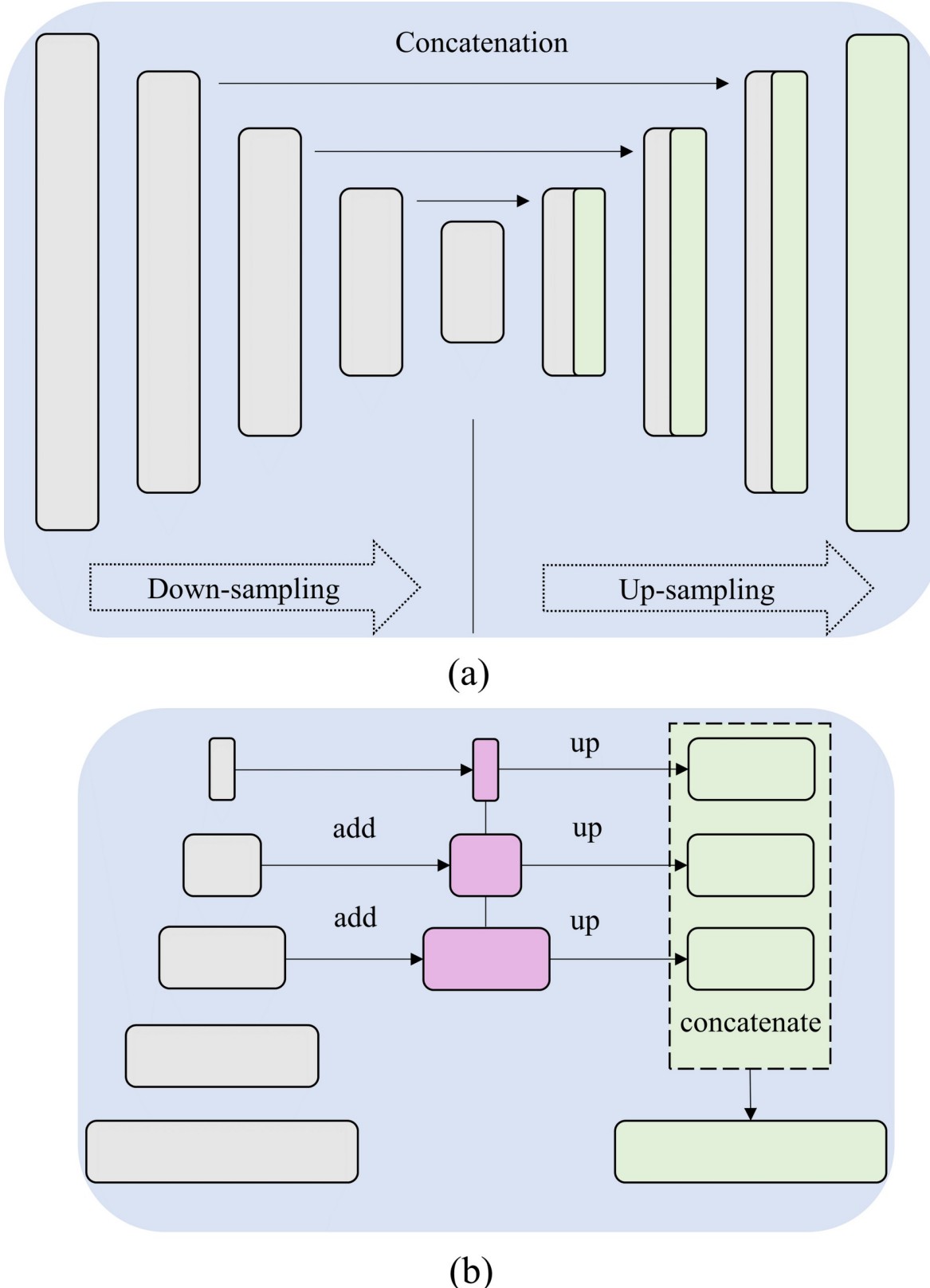

**Fig 4.** The general architecture of the (a) U-Net and (b) FPN model. The input is a three-channel CXR image. The following encoder-backbones are used in this study (EfficientNet-B0, ResNet-18, SE-ResNet-18, DenseNet-121, Inception-V3, MobileNet-V2, trained on ImageNet dataset).

1 test set. Each of the top-3 performing models predicts a bone-suppressed image for an input CXR. The predicted image by the individual models is divided into sub-blocks of M×M dimensions. The optimal value of M [4, 8, 16, 32, 64, 128, 256] is determined through extensive empirical evaluations. For a given sub-block size and in each sub-block, the following are performed: (i) we measured the MS-SSIM score between the sub-block of the bone-suppressed image predicted by each of the top-3 performing models and the corresponding sub-block in respective ground truth soft-tissue image; (ii) we performed a majority voting for the MS-SSIM score to find that image sub-block with the maximum MS-SSIM score and use it in constructing the final bone-suppressed image. The algorithm below discusses these steps. Fig 5 illustrates the steps involved in constructing the DeBoNet.

```
Algorithm
  Input: Ground-truth bone-suppressed image K of 256×256 resolution
  Bone-suppressed Images I = (I_M1, I_M2, I_M3) of 256×256 resolution from
  M = [M_1, M_2, M_3]; M_1, M_2, M_3 are the top-3 performing bone-suppression
  models
  Image sub-block sizes B = [4, 8, 16, 32, 64, 128, 256]
  Output: Final Bone-suppressed image J
  for each sub-block size B
    for each set of bone-suppressed Images I generated by M_1, M_2, M_3
      for each sub-block in K and I_M1, I_M2, I_M3
        compute MS-SSIM between K and I_M1, K and I_M2, K and I_M3
        perform Majority Voting = Max(MS-SSIM(K and I_M1), MS-SSIM(K and
I_M2), MS-SSIM(K and I_M3))
        choose the sub-block with the maximum MS-SSIM value and put it
in its respective position in the final bone-suppressed image J
```

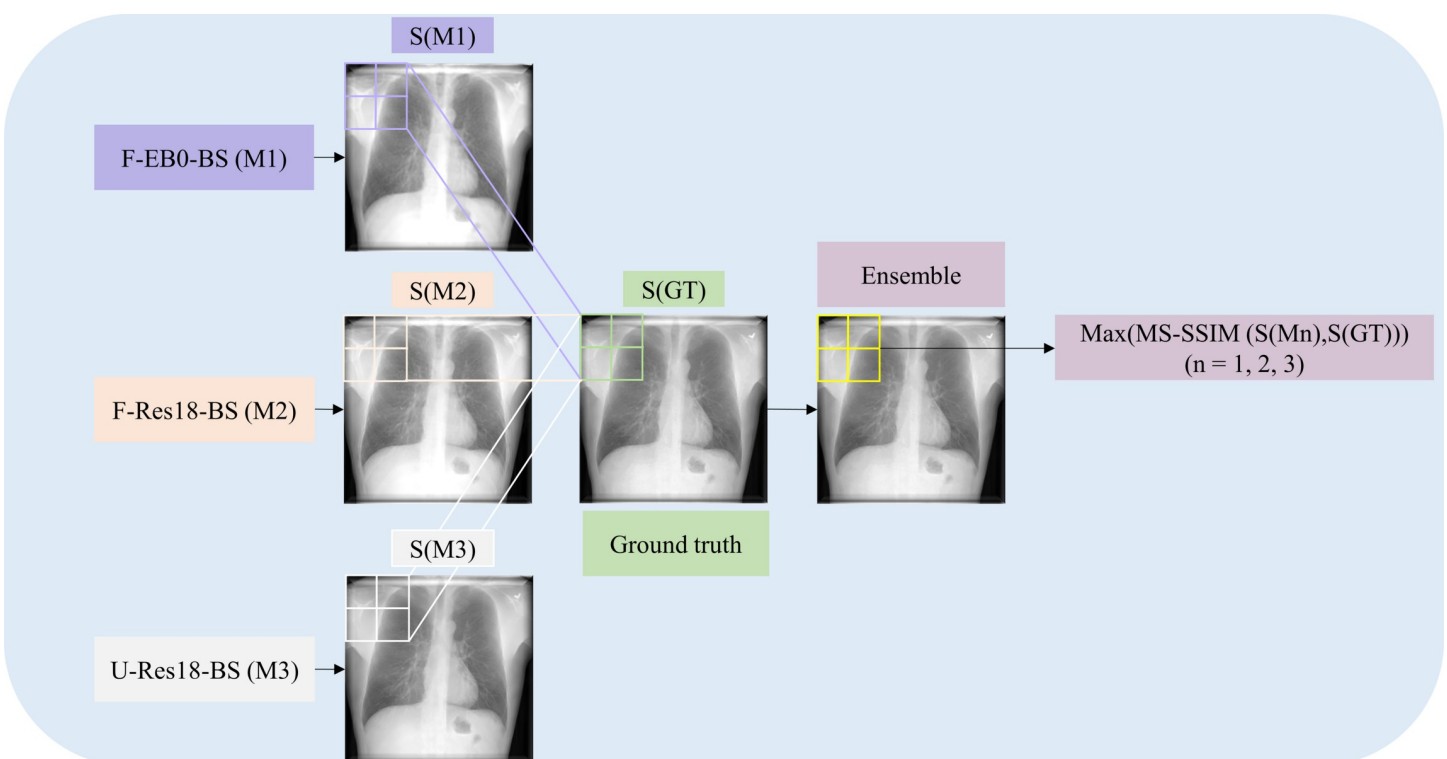

**Fig 5. The architecture of the proposed DeBoNet.**

```
            end for
          end for
        end for
```

**DeBoNet evaluation.** We performed evaluations by using the histograms of the ground truths and the bone-suppressed images predicted by the individual bone-suppression models and the DeBoNet. Several metrics such as correlation, intersection, chi-square distance, and Bhattacharyya distance are measured to investigate for similarity. The higher the value of correlation and intersection, the closer (or more similar) are the histograms of the image pairs. For distance-based metrics such as chi-square and Bhattacharyya, a smaller value indicates a superior match between the histogram pairs. This implies the histograms of the predicted bone-suppressed images closely match their respective ground truths. The mathematical formulations of these metrics can be found in the literature [36]. The average values of the aforementioned metrics are computed for each model and the DeBoNet and compared for statistical significance.

**Classification model.** For classification, we initially used a custom U-Net model proposed in [17] to segment the lung ROI on the CXRs. This approach ensures that the models learn relevant features from the lung ROI and not the surrounding context. The U-Net model is trained to generate 256×256-dimension lung masks. The generated masks are overlaid on the input CXRs to delineate the lung boundaries. The delineated boundaries are cropped to a bounding box containing the lung pixels. The lung-cropped CXRs are further preprocessed to enhance image contrast by clipping the top and bottom 1%, respectively, of all pixel values. We further performed pixel normalization, centering, and standardization to reduce computational complexity during model training.

The encoder of the best-performing bone-suppression model is truncated and appended with the following layers: (i) Zero padding (ZP); (ii) Convolutional layer with 512 filters, each of size 3×3; (iii) Global average pooling (GAP), and (iv) a dense layer with two nodes and Softmax activation to classify the CXRs as showing normal lungs or other findings consistent with COVID-19. This approach is followed to transfer the CXR modality-specific knowledge learned from the bone suppression task to improve performance in a relevant classification task. A study of the literature reveals several works that used CXR modality-specific models to transfer knowledge and improve classification and localization performance in a relevant task [17, 37, 38].

Recall that we use the COVID-19 CXR collection as cases and the RSNA CXR collection as controls for the classification task. Since the ground truth soft-tissue images are not available for these CXRs, the DeBoNet could not be directly used. Instead, the best-performing bone suppression model is selected and applied to these CXR collections. We used 90% of these data for training and 10% for hold-out testing. For consistency, we use a fixed seed and allocated 10% of the training data for validation. The model is then retrained individually on the non-bone-suppressed and bone-suppressed CXR images to classify them as showing no abnormalities or findings consistent with COVID-19. We performed augmentation with random affine transformations such as rotations (-10 to 10 degrees), horizontal and vertical pixel shifting (-5 to 5 pixels), zooming, and horizontal mirroring, to introduce variability into the training process and reduce model overfitting to the training data. The model is compiled using a stochastic gradient descent optimizer with an initial learning rate of 1e-3. The learning rate is reduced whenever the validation performance did not improve. We used callbacks to store model weights and early stopping to prevent overfitting and stored the best weights for further analysis. The best model is used to predict the test set and output class probabilities.

The following metrics are measured to compare model performance: (i) accuracy; (ii) AUROC; iii) precision (P); (iv) recall (R); (v) AUPRC; (vi) F-score; and (vii) MCC. These metrics are expressed below.

$$Accuracy = \frac{TP + TN}{TP + TN + FP + FN} \qquad (2)$$

$$Recall = \frac{TP}{TP + FN} \qquad (3)$$

$$Precision = \frac{TP}{TP + FP} \qquad (4)$$

$$F - score = 2 \times \frac{Precision \times Recall}{Precision + Recall} \qquad (5)$$

$$MCC = \frac{TP \times TN - FP \times FN}{((TP + FP)(TP + FN)(TN + FP)(TN + FN))^{1/2}} \qquad (6)$$

Here, TP, TN, FP, and FN denote the true positive, true negative, false positive, and false negative values, respectively. Additionally, we used our in-house class-selective relevance map (CRM) algorithm [21] to interpret the predictions of the model trained on non-bone-suppressed and bone-suppressed images and ensure they learned to highlight regions containing findings that are consistent with COVID-19.

**Statistical analyses.** We performed statistical analyses to identify the existence of a significant difference in performance achieved by the bone suppression and classification models. For bone suppression, we performed a one-way Analysis of Variance (ANOVA) to analyze if a significant difference existed in the MS-SSIM and chi-square distance values obtained using the top-3 performing bone-suppression models and DeBoNet. We performed Shapiro-Wilk and Levene tests to analyze if the prerequisite conditions of data normality and homogeneity of variances are satisfied to perform one-way ANOVA analyses. For classification, we measured the 95% binomial confidence intervals (CI) as the Exact Clopper-Pearson interval for the MCC metric to compare the classification performance achieved by the models trained on non-bone-suppressed and bone-suppressed images. We used R statistical software (Version 4.1.1) to perform these evaluations.

## Results

### Bone suppression

Recall that the proposed bone suppression models are trained on the augmented NIH-CC-DES-Set 2 dataset and tested using the NIH-CC-DES-Set 1 collection. The performance achieved by the bone suppression models is shown in Table 2. Fig 6 shows the bone-suppressed images predicted using the proposed bone suppression models for an input CXR instance from the test set.

It is observed from Table 2 that the FPN model with the EfficientNet-B0 encoder backbone (F-EB0-BS) demonstrated superior performance for all metrics compared to other models. We observed from Fig 6 that all models predicted bone suppressed images that demonstrated substantial suppression of the bony structures. We further performed a quantitative evaluation to differentiate model performance. In this regard, we observed that the F-EB0-BS model demonstrated the least values for the chi-square and Bhattacharya distances and superior values for

**Table 2. Performance achieved by the proposed bone suppression models using the NIH-CC-DES-Set 1 test set.** The values are given in terms of mean ± standard deviation. The best performances are denoted by bold numerical values in the corresponding columns.

| Model | PSNR | SSIM | MS-SSIM | Correlation | Intersection | Chi-square | Bhattacharya |
|---|---|---|---|---|---|---|---|
| Autoencoder-BS | 33.1861±3.5922 | 0.9371±0.0310 | 0.9798±0.0093 | 0.5949±0.1800 | 8.4827±1.4190 | 1.4279±0.9773 | 0.4009±0.0878 |
| ResNet-BS | 30.9168±3.1286 | 0.9420±0.0261 | 0.9817±0.0092 | 0.5142±0.1831 | 8.2680±1.5036 | 2.6780±1.6202 | 0.4281±0.0884 |
| U-EB0-BS | 35.9098±1.5674 | 0.9359±0.0306 | 0.9795±0.0084 | 0.6529±0.1576 | 8.8000±1.3606 | 0.9004±0.6436 | 0.3813±0.0845 |
| U-Res18-BS | 35.7993±1.4498 | 0.9402±0.0283 | 0.9809±0.0080 | 0.6518±0.1403 | 8.8879±1.4312 | 0.9767±0.4622 | 0.3796±0.0833 |
| U-SE-Res18-BS | 35.531±1.6773 | 0.9325±0.0310 | 0.9773±0.0077 | 0.6421±0.1505 | 8.6794±1.3098 | 1.0484±0.8215 | 0.383±0.0836 |
| U-D121-BS | 33.7751±1.3033 | 0.9284±0.0301 | 0.9746±0.0083 | 0.6017±0.1543 | 8.4233±1.6595 | 1.7434±1.2997 | 0.3852±0.0838 |
| U-IV3-BS | 34.8914±1.7280 | 0.9368±0.0294 | 0.9795±0.0089 | 0.6411±0.1339 | 8.8026±1.4659 | 1.1195±0.4987 | 0.3836±0.0816 |
| U-MobileV2-BS | 27.6842±0.1715 | 0.8593±0.0342 | 0.9136±0.0139 | 0.2583±0.1131 | 5.7133±1.5060 | 10.9967±4.2341 | 0.4704±0.0631 |
| F-EB0-BS | **36.5525±1.6923** | **0.9449±0.0290** | **0.9840±0.0081** | **0.6654±0.1473** | **9.0462±1.4529** | **0.6893±0.4005** | **0.3790±0.0846** |
| F-Res18-BS | 36.3233±1.7004 | 0.9428±0.0281 | 0.9823±0.0079 | 0.6417±0.1424 | 8.8840±1.4194 | 0.9392±0.3799 | 0.3856±0.0833 |
| F-SE-Res18-BS | 36.0318±1.6900 | 0.9418±0.0294 | 0.9821±0.0084 | 0.6334±0.1559 | 8.8531±1.4131 | 1.0227±0.5185 | 0.3853±0.0841 |
| F-D121-BS | 35.2788±1.4938 | 0.9402±0.0283 | 0.9794±0.0082 | 0.6290±0.1365 | 8.7087±1.5015 | 1.2203±0.9092 | 0.3827±0.0818 |
| F-IV3-BS | 33.7446±1.8066 | 0.9369±0.0310 | 0.9793±0.0084 | 0.6225±0.1560 | 8.6645±1.3670 | 1.1846±0.7676 | 0.3910±0.0817 |
| F-MobileV2-BS | 33.5028±1.3452 | 0.9255±0.0320 | 0.9734±0.0088 | 0.5767±0.1743 | 8.1361±1.6643 | 2.3053±1.4224 | 0.3877±0.0844 |

the correlation and intersection measures. Higher values for the correlation and intersection metrics demonstrate that the bone-suppressed images predicted by the F-EB0-BS model closely match that of the ground truth soft-tissue images. Considering the chi-square and Bhattacharyya distance-based metrics, a smaller value indicates a superior match between the images. This signifies that compared to other models, the bone-suppressed image predicted by the F-EB0-BS model closely matches that of the ground truth soft-tissue images. This performance is followed by the FPN model with ResNet-18 encoder backbone (F-Res18-BS) and the U-Net model with the ResNet-18 encoder backbone (U-Res18-BS) that demonstrated markedly improved values for the PSNR, SSIM, MS-SSIM, correlation, intersection, chi-square, and Bhattacharya distance measures compared to other models. These top-3 performing models are further considered to construct the ensemble.

The predicted bone-suppressed images by the top-3 performing models are divided into sub-blocks of M×M dimensions. We empirically determined the value of M [4, 8, 16, 32, 64, 128, 256] that deliver superior bone suppression performance. For a given sub-block size, and in each sub-block, (i) we measured the MS-SSIM score between the sub-block of the bone-suppressed image predicted by each of the top-3 performing models and the corresponding sub-block in the respective ground truth, and (ii) performed a majority voting of the MS-SSIM score for each sub-block to identify the sub-block with the maximum MS-SSIM score and use it in constructing the final bone-suppressed image. Table 3 shows the performance achieved while constructing the DeBoNet using varying sub-block sizes. It is observed from Table 3 that the DeBoNet performance with various sub-block sizes is superior compared to the performance achieved using the top-3 performing models (from Table 2). We observed that using a sub-block size of 4×4, the DeBoNet achieved superior performance in terms of PSNR, SSIM, MS-SSIM, correlation, intersection, chi-square, and Bhattacharya distances compared to using other sub-block sizes and the top-3 performing models. Curiously, we also note a relatively high performance at 256x256 grid dimensions. Studying the correlation between granularity and MS-SSIM score is left as future work.

We performed a one-way ANOVA analysis to observe if a statistically significant difference existed in the MS-SSIM and chi-square values obtained using DeBoNet with sub-block size 4×4, and the top-3 performing bone-suppression models namely, the F-EB0-BS, F-Res18-BS,

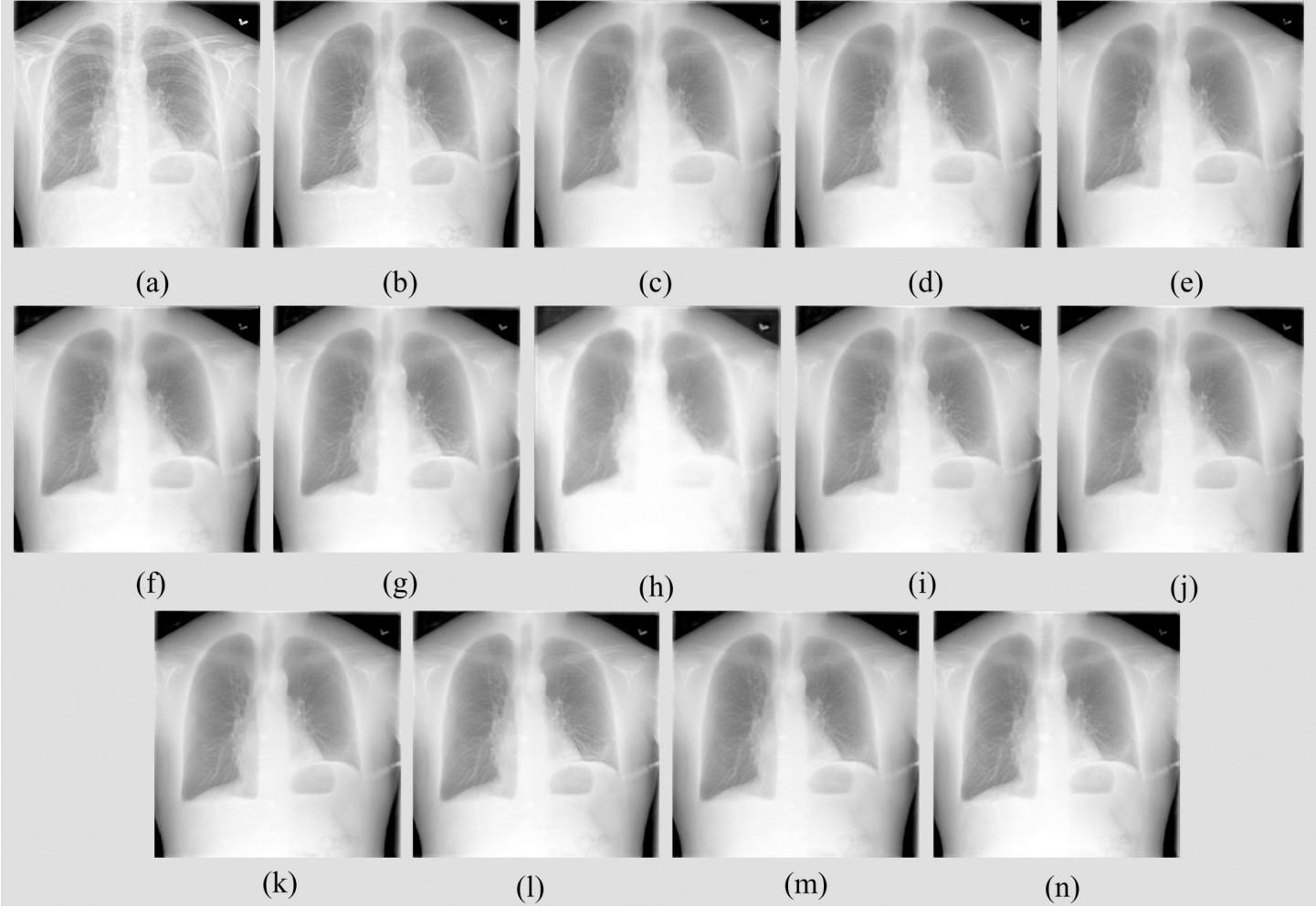

**Fig 6. Bone-suppressed CXR images predicted by the proposed models using a CXR sample from the NIH–CC DES-Set 1 test set.** (a) Original CXR; (b) Ground truth soft tissue image; (c) U-EB0-BS; (d) U-Res18-BS; (e) U-SE-Res18-BS; (f) U-D121-BS; (g) U-IV3-BS; (h) U-MobileV2-BS; (i) F-EB0-BS; (j) F-Res18-BS; (k) F-SE-Res18-BS; (l) F-D121-BS; (m) F-IV3-BS and (n) F-MobileV2-BS.

and U-Res18-BS models. Fig 7 shows the mean plots for the MS-SSIM and chi-square values, respectively, obtained by the models.

The one-way ANOVA analyses require that the assumptions regarding the normal distribution of the data and homogeneity of data variances are satisfied. We performed the

**Table 3. Performance achieved by the DeBoNet using various sizes for the sub-blocks.** The values are given in terms of mean ± standard deviation. The best performances are denoted by bold numerical values in the corresponding columns.

| Block size | PSNR | SSIM | MS-SSIM | Correlation | Intersection | Chi-Square | Bhattacharya |
|---|---|---|---|---|---|---|---|
| 4×4 | **36.7977±1.6207** | **0.9465±0.0272** | **0.9848±0.0073** | **0.6720±0.1404** | **9.0862±1.4413** | **0.6174±0.2726** | **0.3778±0.0839** |
| 8×8 | 36.4574±1.4724 | 0.9226±0.0255 | 0.8721±0.0223 | 0.6344±0.1361 | 8.5230±1.3419 | 1.2636±0.4771 | 0.3806±0.0822 |
| 16×16 | 36.7651±1.6012 | 0.9437±0.0256 | 0.9837±0.0073 | 0.6598±0.1431 | 9.0193±1.4464 | 0.7282±0.3169 | 0.3800±0.0839 |
| 32×32 | 35.7137±1.2588 | 0.8965±0.0266 | 0.8390±0.0237 | 0.5161±0.1218 | 7.1297±1.1228 | 3.4226±1.0079 | 0.3901±0.0801 |
| 64×64 | 36.2657±1.4698 | 0.9218±0.0272 | 0.8719±0.0221 | 0.6297±0.1409 | 8.4949±1.3528 | 1.3402±0.5874 | 0.3815±0.0834 |
| 128×128 | 36.4872±1.5982 | 0.9380±0.0282 | 0.9213±0.0174 | 0.6667±0.1483 | 8.9424±1.4365 | 0.7807±0.4931 | 0.3784±0.0850 |
| 256×256 | 36.5787±1.6885 | 0.9458±0.0284 | 0.9841±0.0080 | 0.6641±0.1470 | 9.0304±1.4599 | 0.7026±0.4129 | 0.3796±0.0849 |

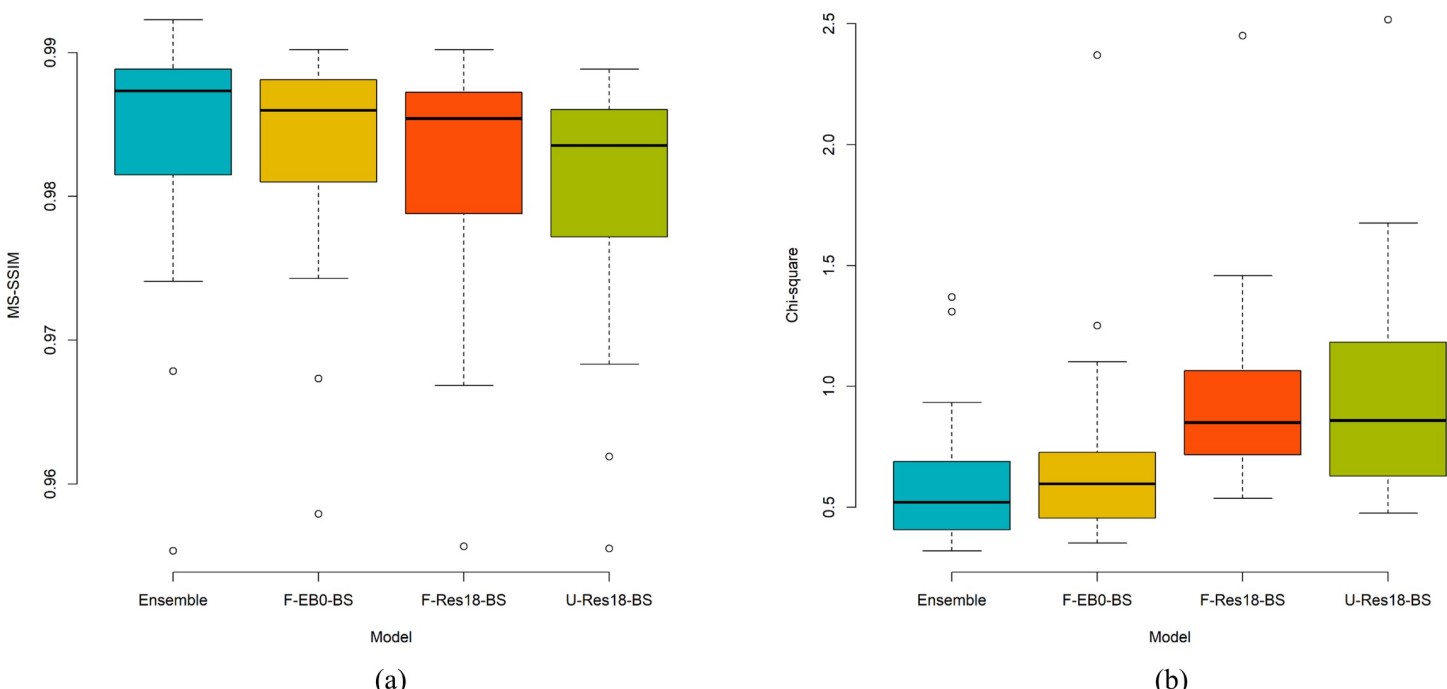

**Fig 7. Statistical analyses using one-way ANOVA.** (a) and (b) shows the mean plot for the MS-SSIM and chi-square values, respectively, obtained by the DeBoNet (4×4), F-EB0-BS, F-Res18-BS, and U-Res18-BS models.

Shapiro-Wilk normality test and Levene test for analyzing the homogeneity of variances. For the MS-SSIM metric, we observed that the $p$-values for the Levene ($p = 0.9828$) and Shapiro-Wilk ($p = 0.3824$) tests are not statistically significant ($p > 0.05$). This confirms that the assumptions of data normality and homogeneous variances are satisfied. Hence, we performed one-way ANOVA analyses by measuring the size of the group, the variance within groups, and the variance between the means of the groups. This information is collectively used to measure the F statistic. In this study, we have four groups/models (i.e., the 4×4 DeBoNet, F-EB0-BS, F-Res18-BS, and U-Res18-BS models) with 27 observations (images) each, hence the distribution is given as F (3, 104). Considering the MS-SSIM metric, we observed that no statistically significant difference existed between the 4×4 DeBoNet and the top-3 performing models (F (3, 104) = 0.886, $p = 0.451$, $p > 0.05$). A similar analysis is performed using the chi-square distance metric. We observed that the conditions of data normality and homogeneous variances are satisfied based on the $p$-values obtained using the Shapiro-Wilk ($p = 0.4768$) and Levene ($p = 0.4321$) tests ($p > 0.05$). The one-way ANOVA analysis revealed that a statistically significant difference existed in the chi-square values obtained using the 4×4 DeBoNet, F-EB0-BS, F-Res18-BS, and U-Res18-BS models (F (3, 104) = 5.838, $p = 0.001$, $p < 0.05$). We further performed Tukey post-hoc analyses to identify the models that demonstrate these significant differences in the chi-square values. We observed that the chi-square distance value obtained using the 4×4 DeBoNet (0.6174±0.2726) is significantly smaller compared to the F-Res18-BS (0.9392±0.3799, $p = 0.0142$) and U-Res18-BS (0.9767±0.4622, $p = 0.0047$) models. These evaluations underscored the fact that the 4×4 DeBoNet achieved significantly smaller values for the chi-square metrics ($p < 0.05$). Also, the chi-square value obtained using the F-EB0-BS model is significantly smaller ($p = 0.0355$) compared to the U-Res18-BS model. Unlike the top-3 performing models, the bone-suppressed images predicted by the 4×4 DeBoNet closely resembled the ground truth soft-tissue images.

Recall that the best-performing F-EB0-BS bone suppression model is used to suppress the bones in the CXRs used in this classification task. This is because the ground truth soft-tissue images are not available for these CXRs. Hence, DeBoNet could not be used. Fig 8 shows the bone-suppressed images predicted by the F-EB0-BS model for instances of CXRs showing findings that are consistent with COVID-19. Note that the F-EB0-BS model generalizes to the unseen CXRs from the classification data that are not used during bone-suppression model training and validation. We observed superior suppression of bones and the image resolution is preserved.

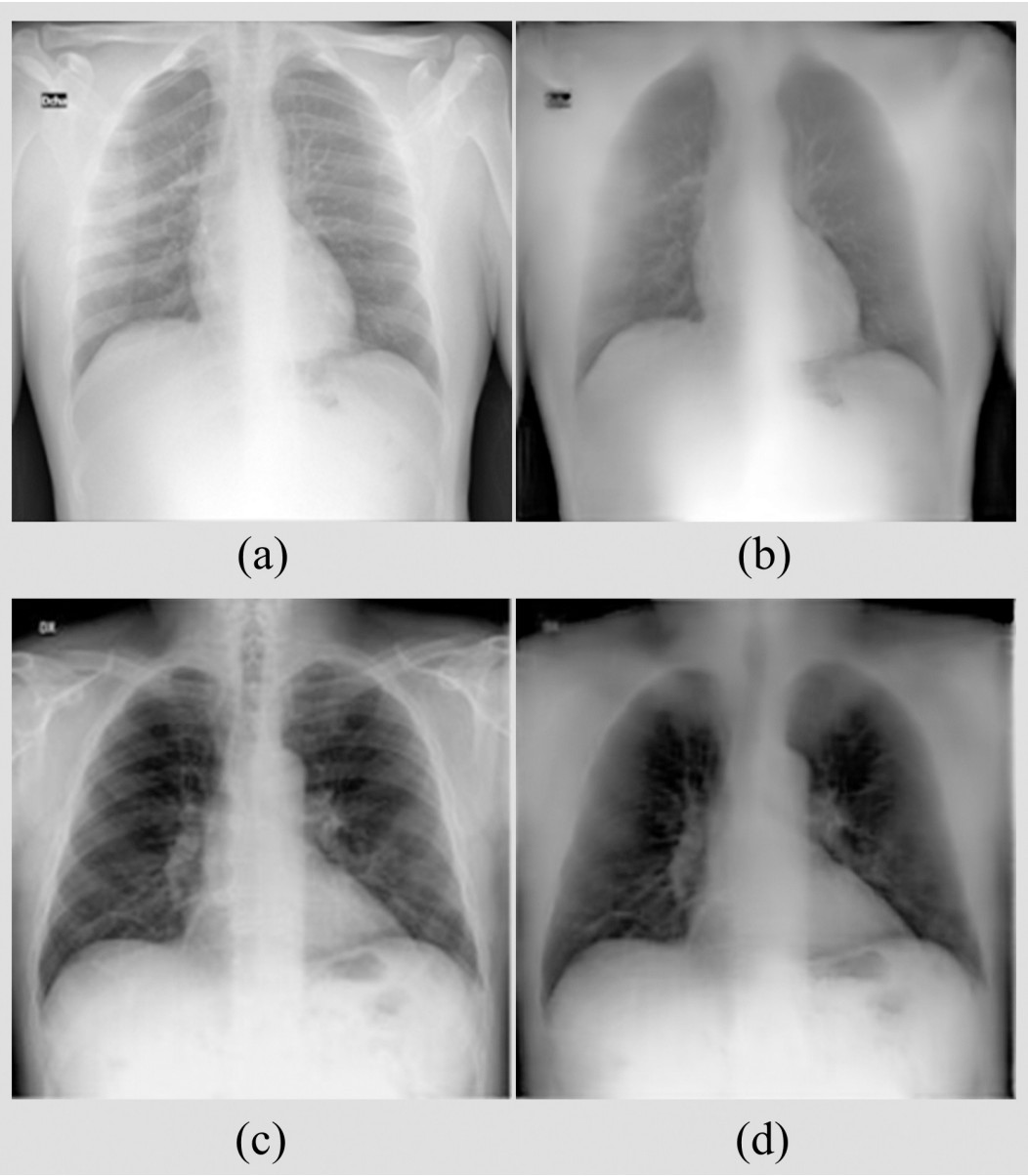

**Fig 8. Bone-suppressed images predicted by the F-EB0-BS model using instances of CXRs with COVID-19-consistent findings.** (a) CXR from the BIMCV-COVID19+ CXR data collection; (b) Corresponding bone-suppressed image; (c) CXR from the Twitter COVID-19 CXR collection, and (d) Corresponding bone-suppressed image.

## Classification

Recall that the encoder of the best-performing F-EB0-BS bone suppression model is truncated and added with the classification layers to classify the CXRs as showing normal lungs or COVID-19-consistent findings. Such an approach is followed to transfer CXR modality-specific knowledge to improve classification performance. The classification model is retrained on the non-bone-suppressed and bone-suppressed CXR images, and the measured performance is shown in Table 4 and illustrated in Fig 9 in terms of AUROC, confusion matrix, normalized Sankey diagram, and AUPRC curves.

We observed from Table 4 and Fig 9 that the classification model trained on bone-suppressed images demonstrated superior performance in terms of accuracy, AUROC, AUPRC, sensitivity, precision, F-score, and MCC metrics, compared to the model trained on non-bone-suppressed images. The 95% binomial CI value obtained for the MCC metric using the model trained on bone-suppressed images demonstrated a tighter error margin, higher precision, and is found to be significantly superior ($p < 0.05$) compared to the MCC metric achieved by the model trained on non-bone-suppressed images.

We qualitatively evaluated the performance of the models trained on non-bone-suppressed and bone-suppressed images to ensure if the models learned to highlight regions containing COVID-19-consistent findings and not the surrounding context. We used the CRM localization tool to interpret model behavior. Fig 10 shows the instances of CXRs, and the CRM-based disease ROI localization obtained using the trained models.

Fig 10A, 10D and 10G show instances of CXRs from the Twitter COVID-19 CXR collection with expert annotations shown in blue bounding boxes. Fig 10B, 10E and 10H show the localization achieved using the model trained on non-bone-suppressed images. It could be observed that the model is highlighting the surrounding context but not COVID-19-consistent manifestations. This demonstrates that the model has not learned relevant features regarding findings that are consistent with COVID-19. Fig 10C, 10F and 10I show the localization achieved using the model trained on bone-suppressed images. We could observe that this model precisely highlighted regions specific to findings that are consistent with COVID-19, thereby demonstrating that the model learned task-specific features, confirming the experts' knowledge about the disease.

## Discussion and conclusions

The observations made from this study underscores the need for (i) customizing a model for the problem under study, (ii) constructing a model ensemble for bone suppression, and (iii) interpreting model behavior.

Our proposed approach facilitates predicting a bone-suppression image given an input CXR image. This is more computationally effective than other studies proposed in the literature [5–9] that propose a series of steps to generate bone-only images and subtract them from input CXRs to increase soft-tissue visibility. A limitation of this approach proposed in the literature is that a sub-optimal generation of bone-only images introduces noise and distortion into the process and may adversely impact decision-making. We proposed several custom

**Table 4. Classification performance achieved with the model trained on non-bone-suppressed and bone-suppressed images.** Data in parenthesis denote the 95% binomial CI measured as the Exact Clopper Pearson interval for the MCC metric. Bold numerical values denote superior performance in respective columns.

| Data | Accuracy | AUROC | AUPRC | Sensitivity | Precision | F-score | MCC |
|---|---|---|---|---|---|---|---|
| Non bone suppressed | 0.8964 | 0.9470 | 0.9275 | 0.8964 | 0.8997 | 0.8962 | 0.7961 (0.7667, 0.8255) |
| Bone suppressed | **0.9820** | **0.9980** | **0.9981** | **0.9820** | **0.9825** | **0.9820** | **0.9645 (0.9510, 0.9780)** |

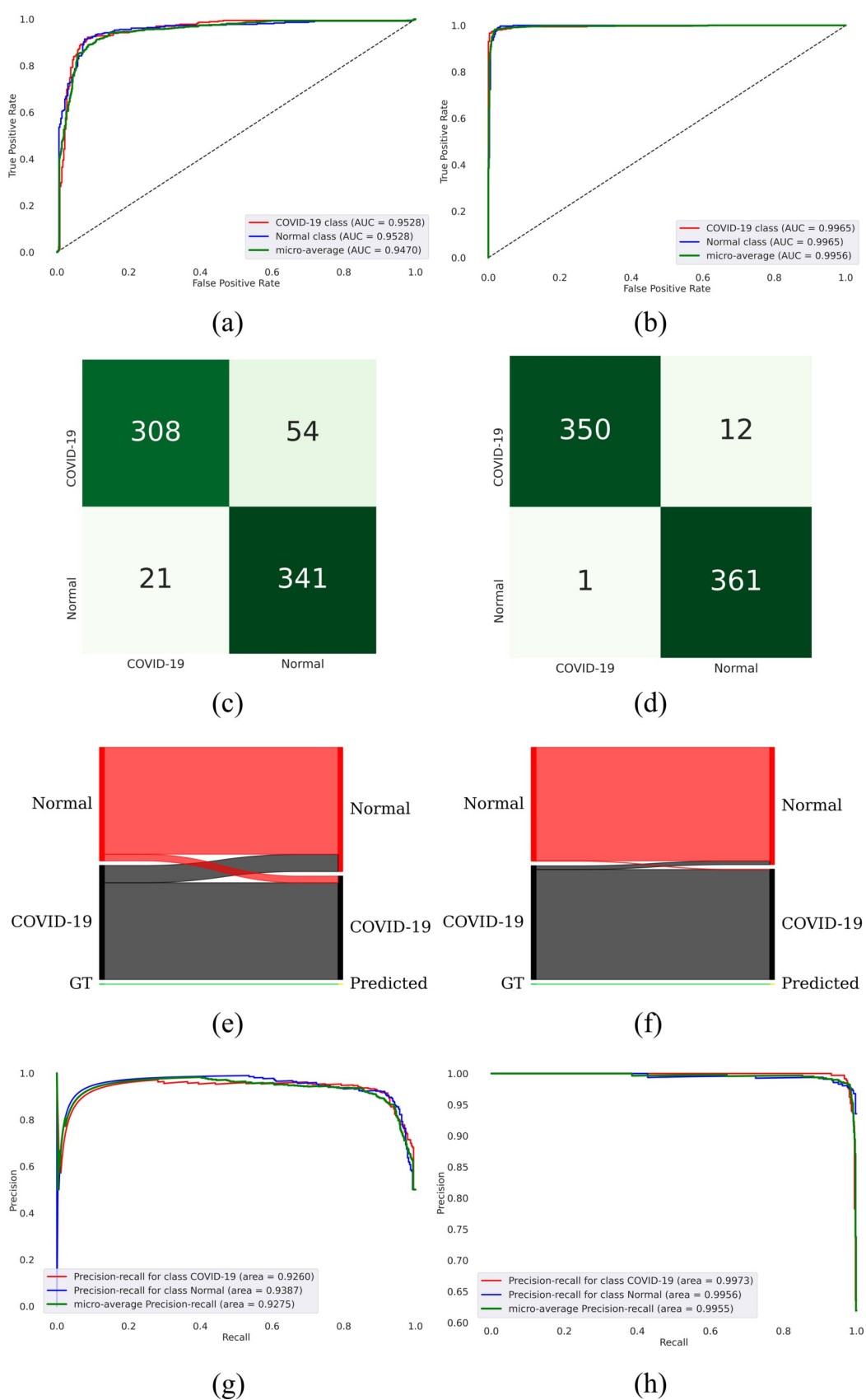

**Fig 9. Classification performance achieved by the model trained on non-bone-suppressed and bone-suppressed images.** (a), (c), (e), and (g) denote the AUROC, confusion matrix, Sankey diagram, and AUPRC curves achieved through training the model with non-bone-suppressed images; (b), (d), (f), and (h) denote the AUROC, confusion matrix, Sankey diagram, and AUPRC curves achieved through training with the bone-suppressed images.

models and experimented with state-of-the-art architectures like U-Nets and FPN using various ImageNet-pretrained encoder backbones to obtain superior bone suppression performance. To the best of our knowledge, this study is the first to explore the use of these models in the context of an image denoising problem where the bony structures in an input CXR are considered noise. Through extensive empirical evaluations, we observed that the FPN model with the EfficientNet-B0 encoder backbone delivered superior bone suppression performance, followed by the FPN model with ResNet-18, and U-Net with ResNet-18 encoder backbones. The bone-suppressed images predicted by these top-3 models appeared sharp while preserving soft-tissue characteristics. Therefore, these images could be used for further CXR image analysis such as screening for cardiopulmonary diseases. We propose an ensemble approach toward bone suppression, called DeBoNet, that demonstrated superior values for PSNR, SSIM, MS-SSIM, correlation, intersection, chi-square, and Bhattacharya distance metrics compared to the individual constituent models. This underscores the fact that the DeBoNet improved bone suppression performance so that the predicted bone suppressed image closely matched the ground-truth, soft-tissue image.

We observed the effect of bone suppression toward improving COVID-19 detection using CXRs. We observed that the classification model trained using bone-suppressed images demonstrated significantly superior performance in terms of accuracy, AUPRC, AUROC, precision, recall, F-score, and MCC, compared to the model trained on non-bone-suppressed images. We further observed through localization studies that the models trained on bone-suppression images precisely highlighted regions showing findings that are consistent with COVID-19, confirming the expert knowledge of the disease. This underscores the fact that, unlike the model trained on non-bone-suppressed images, the models trained on bone-suppressed images learned task-specific features and not the surrounding context, to classify the CXRs to their respective classes. The models trained on non-bone-suppressed images are accurate, however, they demonstrated sub-optimal localization. This underscores the fact that (i) the disease-specific ROI localization ability of a trained model is not related to its classification accuracy and (ii) localization studies are therefore indispensable to interpret the learned behavior of the trained models.

This study suffers from the following limitations: (i) We used the best-performing bone-suppression model, and not the DeBoNet, to suppress bones in the CXR data used for the classification task. This is because we do not have the ground truth soft-tissue images for these CXRs. However, on a positive note, DeBoNet ensemble training helps develop and identify the best performing individual model. (ii) The lack of large-scale publicly available DES CXR datasets is a significant limitation in training the bone-suppression models. The studies reported in the literature [5–9] used the JSRT CXR images and their bone-suppressed counterparts generated by an automated algorithm developed by the researchers from the Budapest University of Technology and Economics [39] to train the bone suppression models. However, these automated algorithms might have introduced noise and artifacts into the bone suppression process, thereby leading to sub-optimal model training and inference. To the best of our knowledge, this is the first study to use DES CXRs to train the bone-suppression models. However, this is not large-scale data and hence may not encompass a wide range of variability in the bone structures. With the increased availability of DES CXRs that would introduce sufficient data diversity into the training process, it would be possible to propose deeper architectures and

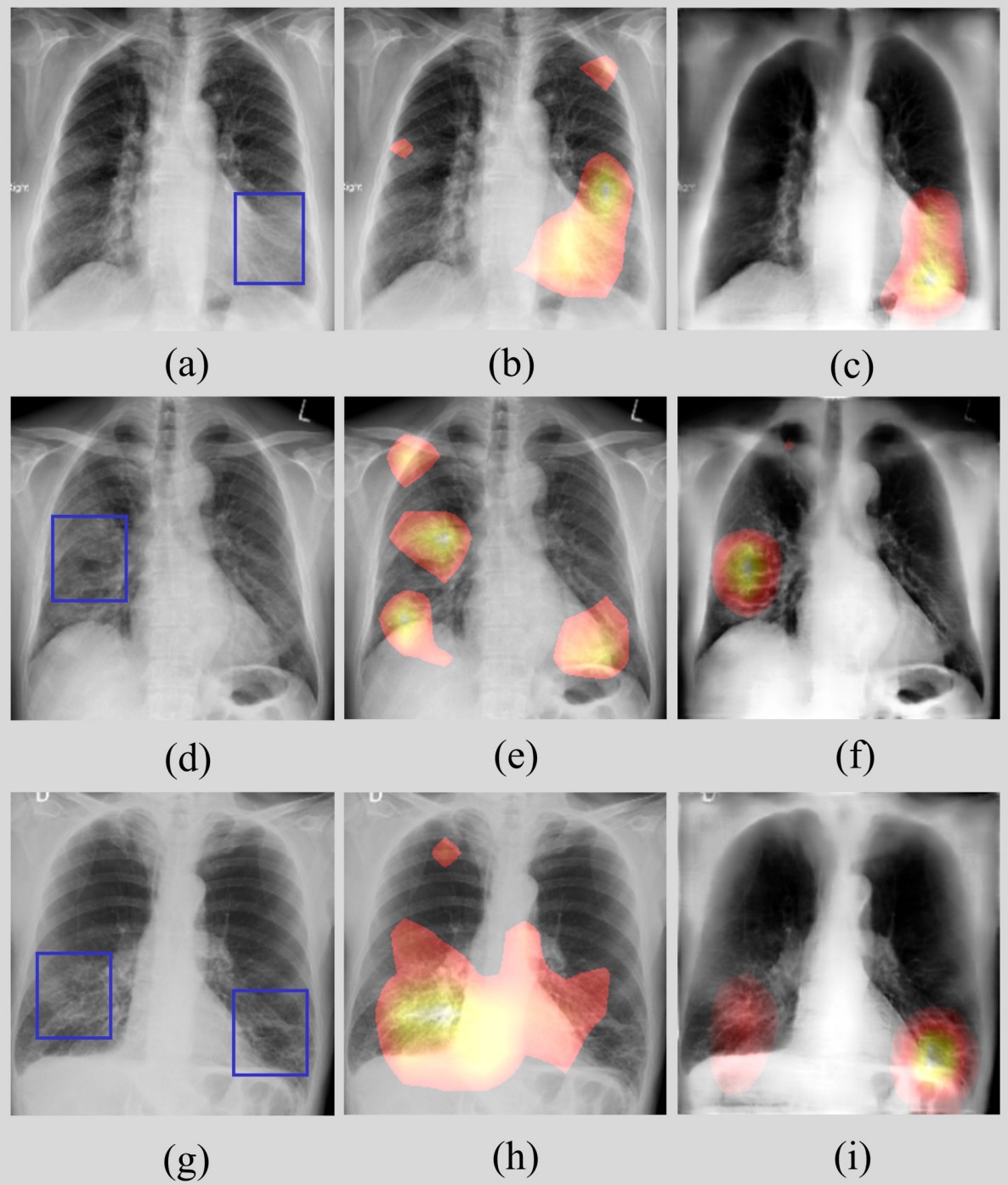

**Fig 10. CRM-based localization of COVID-19-consistent manifestations.** (a) (d) and (g) denote instances of CXRs from the Twitter COVID-19 CXR collection showing COVID-19-manifestations with expert annotations (shown with blue bounding boxes); (b) (e) and (h) shows the regions highlighted by the model trained on non-bone-suppressed images; (c) (f) and (i) shows the COVID-19-consistent ROIs highlighted by the model trained on bone-suppressed images.

improve model confidence, performance, and generalization to real-world data. (iii) This is not a classification-related study, but we wanted to evaluate if bone suppression would

improve performance toward COVID-19 detection. We observed that the model trained on bone-suppressed CXRs improved detection of findings that are consistent with COVID-19, signifying that CXR bone suppression improved the model sensitivity toward COVID-19 classification and localization. Empirically determining the best classification model is outside the scope of this study.

The proposed approach could be extended to other image denoising problems. The importance of using bone suppressed CXRs for detecting other cardiopulmonary abnormalities including lung nodules, TB, pneumothorax, among others would be good research avenues. We believe our results will improve human visual interpretation of COVID-19-consistent findings, as well as automated detection in AI-driven workflows.

## Author Contributions

**Conceptualization:** Sivaramakrishnan Rajaraman, Sameer Antani.

**Data curation:** Gregg Cohen, Lillian Spear, Les Folio.

**Formal analysis:** Sivaramakrishnan Rajaraman.

**Funding acquisition:** Sameer Antani.

**Investigation:** Sivaramakrishnan Rajaraman, Sameer Antani.

**Methodology:** Sivaramakrishnan Rajaraman.

**Project administration:** Sameer Antani.

**Resources:** Gregg Cohen, Les Folio, Sameer Antani.

**Software:** Sivaramakrishnan Rajaraman.

**Supervision:** Sameer Antani.

**Validation:** Sivaramakrishnan Rajaraman.

**Visualization:** Sivaramakrishnan Rajaraman.

**Writing – original draft:** Sivaramakrishnan Rajaraman, Sameer Antani.

**Writing – review & editing:** Sivaramakrishnan Rajaraman, Gregg Cohen, Lillian Spear, Les Folio, Sameer Antani.

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
