## [Decision Letter · Decision Letter 0]

16 Feb 2022

PONE-D-21-38420DeBoNet: A Deep Bone Suppression Model Ensemble to Improve Disease Detection in Chest RadiographsPLOS ONE

Dear Dr. Rajaraman,

Thank you for submitting your manuscript to PLOS ONE. After careful consideration, we feel that it has merit but does not fully meet PLOS ONE’s publication criteria as it currently stands. Therefore, we invite you to submit a revised version of the manuscript that addresses the points raised during the review process.

Please revise the paper by considering the reviewer's comments.

We look forward to receiving your revised manuscript.

Kind regards,

Jie Zhang

Academic Editor

PLOS ONE

Journal Requirements:

“This research was supported by the Intramural Research Program of the National Library of Medicine, and the Clinical Center, both parts of the National Institutes of Health.”

Reviewers' comments:

Reviewer's Responses to Questions

**Comments to the Author**

1. Is the manuscript technically sound, and do the data support the conclusions?

Reviewer #1: Yes

2. Has the statistical analysis been performed appropriately and rigorously? 

Reviewer #1: Yes

3. Have the authors made all data underlying the findings in their manuscript fully available?

Reviewer #1: No

4. Is the manuscript presented in an intelligible fashion and written in standard English?

Reviewer #1: Yes

5. Review Comments to the Author

Reviewer #1: 1. What is the main substantial case in your article?

2. What is the originality of your topic in terms of scientific research and related novel topics?

3. The plagiarism rate is high in your article, reaching 36%. You have to reduce the plagiarism rate according to the permissible limit.

4. What would be required to make your new methodology is their case credible and applicable?

5. What did your topic add to the topic area compared to other published materials?

6. What is the main question covered by your article? Is it relevant and interesting?

7. What does your article add to the subject area compared with other published material?

8. The paper is well written, but it contains many grammatical errors and needs more proofreading.

9. The texts is clear and easy to read.

10. Do the conclusions you have drawn agree with the evidence and arguments presented in your research?

11. Page 12, only written (Algorithm), write Algorithm and its declaration of this algorithm.

12. How many images and how many types did you take to train your network? And What is the deep details of your network( No. of Input Layer, No. of Hidden Layer, …)

13. Was the main question posed answered in your article on which the research was built?

14. Is the tables and figures in your article are sufficient to present your results in a sound, clear and understandable manner?

15. What is the main parameter on the basis of which the entered images were classified?

16. From your point of view, do you think that your article (Including methodology) is original so that it can be cited by the other researchers based on the findings you have taken?

17. Is it possible to develop your research methodology to include other parts of the human body and is not limited to the chest area? And how?

18. Your research paper is serious and worth reading.

• Does the title properly reflect the subject of the paper?

• Does the abstract provide an accessible summary of the paper?

• Do the keywords accurately reflect the content?

• Is the paper an appropriate length?

• Are the key messages short, accurate and clear?

6. PLOS authors have the option to publish the peer review history of their article (what does this mean?). If published, this will include your full peer review and any attached files.

Reviewer #1: No

---

## [Author Response · Author response to Decision Letter 0]

3 Mar 2022

Response to the Editor:

We render our sincere thanks to the Academic Editor for arranging peer review and encouraging resubmission of our manuscript. To the best of our knowledge and belief, we have addressed the concerns of the Academic Editor and the reviewers in the revised manuscript.

Q1: When submitting your revision, we need you to address these additional requirements. 1. Please ensure that your manuscript meets PLOS ONE's style requirements, including those for file naming. The PLOS ONE style templates can be found at https://journals.plos.org/plosone/s/file?id=wjVg/PLOSOne_formatting_sample_main_body.pdf, https://journals.plos.org/plosone/s/file?id=ba62/PLOSOne_formatting_sample_title_authors_affiliations.pdf. 

Author response: We have formatted the manuscript per the templates recommended by the Editor. All figures are checked and converted using the PACE tool recommended by PLOS ONE during submission.

Q2: Thank you for stating the following financial disclosure:

“This research was supported by the Intramural Research Program of the National Library of Medicine, and the Clinical Center, both parts of the National Institutes of Health.”

Please state what role the funders took in the study. If the funders had no role, please state: "The funders had no role in study design, data collection, and analysis, decision to publish, or preparation of the manuscript." If this statement is not correct you must amend it as needed. Please include this amended Role of Funder statement in your cover letter; we will change the online submission form on your behalf.

Author response: We agree to the modifications in the financial disclosure per the Editor’s suggestions. The modified financial disclosure is as follows:

“This research was supported by the Intramural Research Program of the National Library of Medicine, and the Clinical Center, both parts of the National Institutes of Health. The funders had no role in study design, data collection, and analysis, decision to publish, or preparation of the manuscript.”

Q3: Please review your reference list to ensure that it is complete and correct. If you have cited papers that have been retracted, please include the rationale for doing so in the manuscript text or remove these references and replace them with relevant current references. Any changes to the reference list should be mentioned in the rebuttal letter that accompanies your revised manuscript. If you need to cite a retracted article, indicate the article’s retracted status in the References list and also include a citation and full reference for the retraction notice. 

Author response: We ensure that the reference list in the revised manuscript is complete and correct. We have not cited any retracted papers. 

Response to Reviewer #1:

Q1: What is the main substantial case in your article?

Author response: We successfully demonstrate a novel deep learning-based ensemble method for bone suppression in non-dual energy (i.e., conventional) digital chest radiographs CXRs. While existing methods apply multiple steps for bone suppression that may introduce noise and artifacts into the process, the proposed deep learning-based method is trained on a small subset of dual-energy CXRs to recognize bony structures and directly suppresses them in the conventional CXRs (without need for corresponding dual-energy image). This allows us to suppress bones in any conventional CXR. We demonstrate the effectiveness of our bone-suppression technique (named DeBoNet) through testing for gain in performance in detecting pulmonary abnormalities consistent with COVID-19 disease. We observed that the model trained on bone-suppressed CXRs significantly outperformed the model trained on non-bone-suppressed images. 

Q2: What is the originality of your topic in terms of scientific research and related novel topics? 

Author response: The originality of the proposal lies in producing a new knowledge toward bone suppression in conventional chest radiographs without the need for corresponding dual-energy images. This method, called DeBoNet, would enable researchers worldwide to analyze hundreds of thousands of CXRs that are publicly available for cardiopulmonary diseases without interference from bony structures. We experimented, observed, and discussed the efficacy of our approach to solving the problem of bone suppression, interpreted, and analyzed significance in the reported results. We also discussed the current limitations and the score for future research in bone suppression. 

Q3: The plagiarism rate is high in your article, reaching 36%. You have to reduce the plagiarism rate according to the permissible limit. 

Author response: We would like to kindly let you know that our article is available as an arXiv preprint at https://arxiv.org/abs/2111.03404. This might be a potential reason for a high plagiarism rate. However, we observed that PLOS ONE encourages the authors to share their research on a preprint server before submission. We do not have any “recycled” material in this novel work.

Q4: What would be required to make your new methodology is their case credible and applicable? 

Author response: The credibility of this research stems from several facts including the rationale for the study, the process of data collection, novel methodology, the observed results and analysis, and subsequent claims. The trustworthiness of this proposal lies in using the most appropriate data for analysis (using the dual-energy radiographic image ground truth compared to the literature using data from automated bone suppression methods) and performing significance analysis using the reported results, ensuring that it is not reported by chance. We demonstrate a performance gain in bone suppression and also an improvement in detecting pulmonary abnormalities that are consistent with COVID-19 disease using the bone-suppressed chest radiographs. As well, the proposed approach could be extended to other image denoising problems. Further, we have made the code publicly available to support reproducibility. Link is available in the revised manuscript and will become active upon publication of the manuscript. 

Q5: What did your topic add to the topic area compared to other published materials? 

Author response: A study of the literature reveals several works published on suppressing bones in CXRs. These studies involve using (i) commercial software, (ii) conventional machine learning methods using hand-crafted feature descriptors, or (iii) state-of-the-art deep learning (DL) models to initially generate bone-only images and further subtract them from the original CXR to increase soft-tissue visibility. These studies, in general, propose multiple steps to generate bone-only images and subtract them from the original CXRs to increase soft-tissue visibility. A limitation of this approach is that an inaccurate generation of bone-only images would lead to introducing noise, reducing the visibility of soft tissues, increasing interpretation errors, and adversely impacting decision-making. As of the writing of this manuscript and to the best of our knowledge, there are no other articles in the literature that propose an automated method to generate a soft-tissue image directly from the original CXR image, alleviating the need for intermediate bone image generation and subsequent subtraction methods. We propose a deep learning-based method that is trained on dual-energy CXRs to recognize bony artifacts and directly suppress them in the conventional CXRs. We further demonstrate the impact of bone suppression by evaluating the gain in performance in detecting pulmonary abnormality consistent with COVID-19 disease. To our best knowledge, this is the first study to perform such analyses. 

Q6: What is the main question covered by your article? Is it relevant and interesting?

Author response: This study aims to improve bone suppression in chest radiographs. We experimented with individual models and their ensembles and observed that the ensemble of the top-3 performing models resulted in improve bone suppression compared to individual models and other ensembles. We further demonstrate the impact of bone suppression by evaluating the gain in performance in detecting pulmonary abnormality consistent with COVID-19 disease. We believe the proposal is relevant to chest X-ray analysis and would interest the readers for this is the first study to perform such analyses.

Q7: What does your article add to the subject area compared with other published material? 

Author response: We wish to reiterate our response to Q5.

Q8: The paper is well written, but it contains many grammatical errors and needs more proofreading. The text is clear and easy to read.

Author response: Thanks for your appreciative comments. The revised manuscript has been proofread by a native English speaker and corrected for typos and grammatical errors. 

Q9: Do the conclusions you have drawn agree with the evidence and arguments presented in your research? 

Author response: Yes. We identified the gaps in the literature and proposed a novel bone suppression method to improve performance. We discussed these observations and highlighted key findings in our study. The observations made from this study underscores the need for (i) customizing a model for the problem under study, (ii) constructing a model ensemble for bone suppression, and (iii) interpreting model behavior. Such discussion would help advance the understanding of the problem of bone suppression. We performed a significance analysis in the reported results, establishing that our observations are well-founded. We also discussed the current limitations of our study. Further, we discussed how the proposed approach could be extended to other applications. We believe our results will improve bone suppression and human visual interpretation of pulmonary abnormalities, as well as automated detection in AI-driven workflows.

Q10: Page 12, only written (Algorithm), write Algorithm and its declaration of this algorithm. 

Author response: The algorithm is discussed in p. 12 of the revised manuscript. However, we also include it below for the reviewer’s convenience:

Algorithm

Input: Ground-truth bone-suppressed image K of 256×256 resolution

Bone-suppressed Images I = (IM1, IM2, IM3) of 256×256 resolution from M = [M1, M2, M3] ; M1, M2, M3 are the top-3 performing bone-suppression models

Image sub-block sizes B = [4, 8, 16, 32, 64, 128, 256]

Output: Final Bone-suppressed image J

for each sub-block size B

for each set of bone-suppressed Images I generated by M1, M2, M3

 for each sub-block in K and IM1, IM2, IM3

 compute MS-SSIM between K and IM1, K and IM2, K and IM3

perform Majority Voting = Max(MS-SSIM(K and IM1), MS-SSIM(K and IM2), MS-SSIM(K and IM3))

choose the sub-block with the maximum MS-SSIM value and put it in its respective position in the final bone-suppressed image J

 end for

 end for

end for

Q11: How many images and how many types did you take to train your network? And What is the deep details of your network( No. of Input Layer, No. of Hidden Layer, …) 

Author response: The details about the number of layers and the number of filters in each layer of the various models proposed in this study are shown in Fig. 2, Fig. 3, and Fig. 4. The details about the input, intermediate, and output layers of these models are discussed in pp. 8 – 11. The training methods are discussed in lines 244 – 257 on p. 11. 

Q12: Was the main question posed answered in your article on which the research was built? 

Author response: Yes. We have answered these questions through our research findings. The methods proposed, the experiments conducted, and subsequent discussions answer our research questions about bone suppression. We have included the findings of our experiments, discussed the results with significance analysis, and explained how these observations answered the research question of bone suppression in chest radiographs. 

Q13: Is the tables and figures in your article are sufficient to present your results in a sound, clear and understandable manner? 

Author response: We believe the tables and figures presented in this study would suffice to convey the observations from this study and help propose novel methods in the future toward bone suppression and subsequent analysis. 

Q14: What is the main parameter on the basis of which the entered images were classified? 

Author response: We believe that the query is about the performance metrics used to evaluate the bone suppression and classification models. The proposed bone-suppression models are trained and evaluated using the following metrics: (i) loss, (ii) PSNR, (iii) SSIM, and (iv) MS-SSIM. The classification models are trained and evaluated using the following metrics: (i) accuracy; (ii) AUROC; iii) precision (P); (iv) recall (R); (v) AUPRC; (vi) F-score; and (vii) MCC.

Q15: From your point of view, do you think that your article (Including methodology) is original so that it can be cited by the other researchers based on the findings you have taken? 

Author response: We affirm that our article is original and can be cited by other researchers based on the important findings reported in our study. This is the first study to successfully demonstrate a novel deep learning-based ensemble method for bone suppression in non-dual energy (i.e., conventional) digital chest radiographs CXRs without the need for a corresponding dual-energy image. The effectiveness of our bone-suppression technique (named DeBoNet) was proven through improvements in disease detection. The code will be shared and will allow researchers to suppress bones in the hundreds of thousands of PA CXRs that are publicly available for research use. 

Q16: Is it possible to develop your research methodology to include other parts of the human body and is not limited to the chest area? And how? 

Author response: The proposed approach could be extended to other image denoising problems. The key requirement is that during training there be a corresponding dual-energy CXR in the training, or if using other modalities a bone-only version is available to train the deep-learning algorithm to recognize the bones. We believe our results will improve human visual interpretation of disease findings, as well as other similar automated detection in AI-driven workflows.

Q17: Your research paper is serious and worth reading.

Author response: We render our sincere thanks to the reviewer for the valuable comments and appreciation of our study. To the best of our knowledge and belief, we have addressed the reviewer’s concerns.

---

## [Editor Report · Decision Letter 1]

7 Mar 2022

DeBoNet: A Deep Bone Suppression Model Ensemble to Improve Disease Detection in Chest Radiographs

PONE-D-21-38420R1

Dear Dr. Rajaraman,

We’re pleased to inform you that your manuscript has been judged scientifically suitable for publication and will be formally accepted for publication once it meets all outstanding technical requirements.

Kind regards,

Jie Zhang

Academic Editor

PLOS ONE
---

## [Editor Report · Acceptance letter]

9 Mar 2022

PONE-D-21-38420R1 

DeBoNet: A Deep Bone Suppression Model Ensemble to Improve Disease Detection in Chest Radiographs 

Dear Dr. Rajaraman:

I'm pleased to inform you that your manuscript has been deemed suitable for publication in PLOS ONE. Congratulations! Your manuscript is now with our production department. 

Kind regards, 

on behalf of

Dr. Jie Zhang 

Academic Editor

PLOS ONE